# Involvement of Anoikis in Dissociated Optic Nerve Fiber Layer Appearance

**DOI:** 10.3390/ijms22041724

**Published:** 2021-02-09

**Authors:** Tsunehiko Ikeda, Kimitoshi Nakamura, Takaki Sato, Teruyo Kida, Hidehiro Oku

**Affiliations:** 1Department of Ophthalmology, Osaka Medical College, Takatsuki-City 569-8686, Osaka, Japan; opt147@osaka-med.ac.jp (T.S.); opt038@osaka-med.ac.jp (T.K.); opt025@osaka-med.ac.jp (H.O.); 2Nakamura Eye Clinic, Matsumoto-City 390-0811, Nagano, Japan; nakamura-ganka01@tuba.ocn.ne.jp

**Keywords:** macular hole (MH), vitrectomy, internal limiting membrane (ILM), dissociated optic nerve fiber layer (DONFL), optical coherence tomography (OCT), anoikis, βA3/A1 crystallin, E-cadherin, retinal ganglion cell

## Abstract

Dissociated optic nerve fiber layer (DONFL) appearance is characterized by dimpling of the fundus when observed after vitrectomy with the internal limiting membrane (ILM) peeling in macular diseases. However, the cause of DONFL remains largely unknown. Optical coherence tomography (OCT) findings have indicated that the nerve fiber layer (NFL) and ganglion cells are likely to have been damaged in patients with DONFL appearance. Since DONFL appearance occurs at a certain postoperative period, it is unlikely to be retinal damage directly caused by ILM peeling because apoptosis occurs at a certain period after tissue damage and/or injury. However, it may be due to ILM peeling-induced apoptosis in the retinal tissue. Anoikis is a type of apoptosis that occurs in anchorage-dependent cells upon detachment of those cells from the surrounding extracellular matrix (i.e., the loss of cell anchorage). The anoikis-related proteins βA3/A1 crystallin and E-cadherin are reportedly expressed in retinal ganglion cells. Thus, we theorize that one possible cause of DONFL appearance is ILM peeling-induced anoikis in retinal ganglion cells.

## 1. Introduction

Dissociated optic nerve fiber layer (DONFL) appearance, a fundus finding first reported by Tadayoni et al. [1] in 2001, is observed after vitrectomy with the internal limiting membrane (ILM) peeling in macular diseases, such as a macular hole (MH) and the epiretinal membrane (ERM), or for rhegmatogenous retinal detachment [2,3]. It has also been reported that the appearance of DONFL can occur after secondary ILM peeling for hematomas under the ILM [4]. Ophthalmoscopically, DONFL appearance features arcuate striae color changes, which appear to be multilinear in shape, that are frequently observed in the macular nerve fiber layer (NFL) (Figure 1A,B). Since the appearance of DONFL is not common in patients who have undergone vitrectomy without ILM peeling [2,3], it is thought that ILM peeling is certainly involved in its occurrence. Reportedly, DONFL appearance commonly occurs at 1–3 months after vitrectomy, and thereafter, persists in some patients or becomes slightly less distinct in other patients [5,6]. Although the incidence of DONFL varies from report to report, it apparently occurs in approximately 30–60% of MHs in which an ILM peel is performed [1,2,3,4,5,6]. Although previous studies have suggested various factors associated with DONFL appearance, its exact cause remains unclear.

## 2. Anatomical Features of DONFL Appearance

While DONFL appearance can be observed by ophthalmoscopy or color fundus photography, more detailed graphic images can be obtained by infrared fundus photography or optical coherence tomography (OCT). Kishimoto et al. [6] reported that examination by OCT is highly likely to detect DONFL appearance even when it cannot be detected on color fundus photographs. Kusuhara et al. [7] and Pichi et al. [8] reported that “en face” OCT layer imaging was useful for the diagnosis of DONFL appearance. Reportedly, when scanned vertically to the arcuate striae of DONFL appearance using OCT, small dimples are observed in the retinal NFL (RNFL), which are consistent with the striae [9]. In patients without DONFL appearance, OCT fails to visualize distinct dimples in the RNFL. In a report by Miura et al. [10], the authors demonstrated using scanning laser ophthalmoscopy (SLO) of an MH that DONFL appearance is a change in the retinal surface. In a report by Mitamura et al. [9], the authors measured the RNFL thickness (i.e., the mean RNFL thickness at both ends of the dimple site) and the depth of the dimples at three sites in patients using OCT, and their findings showed that the mean RNFL thickness at the dimple margins was 38.1 ± 9.3 μm, that the depth of each dimple was 28.6 ± 8.0 μm, and that the depths of all dimples measured were limited to the RNFL thickness. Kim et al. [11] reported that DONFL appearance gradually increased after surgery and that in 57.7% of the cases, DONFL appearance was confined to the NFL at 6-months postoperative, while in 30.8% of the cases, DONFL appearance extended to the inner plexiform layer. It should be noted that Demirel et al. [12] also reported results consistent with those findings.

### 2.1. DONFL Appearance and Visual Function

Several previous studies have investigated the relationship between DONFL appearance and visual function in patients undergoing vitrectomy for an MH, and the findings in those reports often revealed that there were no significant differences in improvement rates of visual acuity between eyes with and without DONFL appearance [1,2,12,13]. It was initially suspected that DONFL appearance might be associated with optic nerve fiber disorders because its morphology resembles RNFL defects observed in glaucoma cases. Tadayoni et al. [1] performed Goldmann dynamic visual field testing in three patients with DONFL appearance, and their findings revealed that there were no visual field abnormalities. Moreover, numerous other studies evaluating visual function using other visual field tests also reported that DONFL appearance is not associated with visual field abnormalities. Mitamura et al. [14] investigated the retinal sensitivity in the area of DONFL appearance using the Micro Perimeter 1 (MP-1; NIDEK Co., Ltd., Gamagori, Japan) device and reported that there was no difference in retinal sensitivity between the DONFL appearance area showing arcuate striae and the area showing a normal color. Similar findings were also reported by Imai et al. [15] and Ito et al. [2], who performed visual function analysis using SLO microperimetry and found that DONFL appearance is due to dehiscence of the optic NFL and is not a defect of the nerve fiber bundles. Accordingly, it has been generally accepted that DONFL appearance is dimpling localized in the RNFL and is unlikely to be nerve fiber injury. However, Demirel et al. [12] reported the observation of significant thinning of the ganglion cell inner plexiform layer and that it was correlated with reduced visual acuity. OCT scans of our study patients also showed frequent thinning and irregularity of the inner retinal layer (Figure 2A,B). Thus, these findings appear to indicate that the NFL and ganglion cells are very likely to have been damaged in patients with DONFL appearance. A possible reason for the lack of marked functional abnormalities is that the visual testing in the abovementioned studies may have failed to detect visual impairment due to a mild degree of tissue damage.

### 2.2. Previous Studies on the Pathogenesis of DONFL Appearance

Since the ILM is the basement membrane of Müller cells, ILM peeling during vitreous surgery may damage Müller cells. Runkle et al. [13] assessed the relationship between DONFL and intraoperative membrane-peeling dynamics as visualized using intraoperative OCT and evaluated the functional implications of DONFL. Their findings showed the acute post-peel increase in inner retinal thickness, and they concluded that one mechanism in the development of DONFL appearance may be intraoperative trauma to the inner retina, potentially during ILM peeling. In regard to this matter, Tadayoni et al. [1] proposed two hypotheses, the first being that the inner surface of the ILM is smooth, but its outer surface is irregular, and thus, DONFL appearance is an irregularity on the retinal surface after ILM peeling, and the second being that DONFL appearance is caused by the cleavage of the nerve fiber bundles due to ILM peeling-associated damage to the Müller cells, which keep the optic nerve fiber bundles close together. Steel et al. [16] described that the extent of DONFL observed postoperatively can be partly explained by the amount of cellular debris on the retinal side of the peeled ILM, while Park et al. [17] reportedly found no difference in the frequency of DONFL on the use of indocyanine green. Kim et al. [11] assessed the association between the appearance of DONFL after ILM peeling and changes in the macular NFL area by spectral-domain OCT. Their findings showed that there was no significant difference in average macular NFL area between group I (presence of DONFL) and group II (absence of DONFL) at 6-months postoperative. Based on these results, they reported that DONFL appearance started to occur at approximately 1-month postoperatively, and was thus unlikely to be mechanical damage to the NFL caused by ILM peeling and that DONFL appearance might represent macular NFL rearrangement and reorganization [5]. Spaide [18] reported that the correlation of the volume-rendered images with B-scan spectral-domain OCT showed focal areas of thinning of the ganglion cell layer with decreased reflectivity from the NFL in the areas of DONFL. They also described that DONFL seems to be the result of an interplay between trauma and healing processes constrained by NFL and does not appear to be due to the dissociation of optic nerve fibers. As Spaide noted, the word “dissociated” may not be correct, and the term “inner retinal dimpling” may be more appropriate. Their findings also reportedly showed that DONFL appearance is the result of an interplay between the damage to Müller cells caused by ILM peeling and the healing processes, and not damage to the RNFL itself. Hisatomi et al. [19] examined retinal changes after vitrectomy with ILM peeling by using a cynomolgus monkey model and focused on surgical damages of ILM peeling for a long observational period of 3 years. Their ultrastructural studies showed that most of the ILM peeling area was covered with glial cells during the wound healing processes and that retinal changes were found comparable with DONFL appearance, which was clinically observed with OCT. However, the mechanism underlying such changes observed 1 month or more after vitreous surgery has yet to be fully elucidated. Moreover, Müller cells span across the entire retina. Thus, if DONFL appearance is ILM peeling-associated damage to the Müller cells, then a localized lesion within the inner sensory retina is not explicable.

## 3. Involvement of Anoikis in the Pathogenesis of DONFL Appearance

There are two types of cells: anchorage-dependent and anchorage-independent cells. Anchorage-dependent cells proliferate while adhering to the culture vessel, and many cells, such as epithelial cells, belong to this category. On the other hand, cultured cells that proliferate while floating in the medium are known as anchorage-independent cells, such as blood cells. The shape of the whole body is maintained by adhesion between cells that constitute the whole body, and these cells are anchorage-dependent. Blood cells dispersed throughout the whole body are anchorage-independent. Anchorage-dependent cells undergo apoptosis when they are detached from proximal cells or the basement membrane and become suspended. This is probably a form of cellular suicide (i.e., programmed cell death) that naturally occurs to prevent the detached cells from migrating to other organs and disrupting the homeostasis of the surrounding tissues. Apoptosis that occurs when anchorage-dependent cells detach from the surrounding extracellular matrix (i.e., the loss of cell anchorage) is termed anoikis [20,21,22]. For many anchorage-dependent cells, adhesion to the extracellular matrix is an essential function/characteristic for survival and proliferation. However, cancer cells reportedly become anoikis resistant, and the resulting anchorage independence is thought to allow cancer cells to evade apoptosis and become metastatic [23,24]. Moreover, anchorage-dependent cells require attachment to solid substrates, such as basement membranes, and this cell adhesion is reportedly mediated by integrins [25]. Once integrin binding to basement membranes is disrupted, cells become suspended. To date, more than 25 types of integrins have been confirmed. When certain types of integrins are released from the substrates, signals that induce anoikis are transmitted to the nucleus through various kinases, such as focal adhesion kinase, thereby leading to anoikis [26]. The types of integrins that trigger anoikis differ according to cell types. Retinal ganglion cells express focal adhesion kinase [27,28], and the expression of integrins has reportedly been detected in the ILM and neural retina [25]. Therefore, ILM peeling may cause integrins to trigger anoikis by transmitting signals that induce anoikis to the nuclei of ganglion cells through focal adhesion kinase.

### 3.1. Anoikis and Eye Diseases

In the field of ophthalmology, a few previous studies have investigated the association between eye disease and anoikis, and anoikis and its involvement in the onset of eye diseases has reportedly been observed in retinal pigment epithelium (RPE) cells in cases of age-related macular degeneration (AMD) [29,30,31], in pericytes in cases of diabetic retinopathy (DR) [32,33], and vascular endothelial cells in cases involving retinal microaneurysms [34]. Ng et al. [29] identified the HTRA1 insertion–deletion (indel) variant (c.34delCinsTCCT) in 195 exudative AMD patients and 390 controls and followed by studying its biological functions in RPE cells. As the result of that, they discovered that ARPE-19 cells expressing HTRA1 c.34delCinsTCCT variant had higher cell viability and lower cell apoptosis and that the cells were less responsive to anoikis as a mechanism involved in AMD. Petrovski et al. [31] described that the model using ARPE-19 cells can be used for studying both dry and wet types of AMD in vitro and for testing different pharmacological aspects affecting AMD. Liu et al. [32] reported that cysteine-rich protein 61 (Cyr61) and connective tissue growth factor (CTGF), which are an anoikis-prone phenotype to rat retinal pericytes, are aberrantly expressed in the retinal vasculature from the early stages of DR. Yang et al. [33] reported that the increased matrix metalloproteinase-2 (MMP-2) activity compromises retinal pericyte survival possibly through MMP-2 action on extracellular matrix (ECM) proteins and/or direct association of MMP-2 with integrins, which promotes apoptosis/anoikis by loss of cell contact with an appropriate ECM. López-Luppo et al. [34] described that the senescent endothelial cells were discontinuously covered by fibronectin, and p16 colocalized with the β1 subunit of fibronectin receptor α5β1 integrin under the endothelial cellular membrane, thus suggesting anoikis as a mechanism involved in endothelial cell apoptosis during the formation of retinal microaneurysms.

In all such cases, basement membranes involved in the survival of these cells were reportedly subjected to some form of damage, which could be considered to lead to anoikis. Anoikis is a biological phenomenon that occurs upon detachment of anchorage-dependent cells from the substrates, and it has been reported that the likelihood of anoikis occurrence varies depending on cell type.

There are other reports of anoikis being involved in cell death in the central nervous system. In the hippocampus of the brain, it has been reported that neurogenesis occurs as well as neuronal death caused by anoikis [35]. Anoikis has also been reported to be involved in the development of degenerative diseases in the brain, such as Alzheimer’s disease, Parkinson’s disease, and Wallerian degeneration [36,37,38]. These findings suggest that anoikis may be involved in neuronal cell death, not only in the sensory retina but also in various regions.

### 3.2. Anoikis and Glial Cells

As mentioned above, there is a hypothesis that DONFL appearance is due to damage to Müller cells caused by ILM peeling. Previous studies have reported an association of anoikis with the glial cells of astrocytes, oligodendrocytes, and Schwann cells [39,40,41,42,43]. Astrocytes and Müller cells are known to be major types of retinal glial cells, and astrocytes play an important role in the formation of the nervous system, the maintenance of extracellular fluid homeostasis, and the formation of the blood–brain barrier. Astrocytes in the brain project foot processes (i.e., astrocytic endfeet) that envelop blood vessels, neurons, and the pia mater to form the glia limitans. These endfeet cover the external wall of capillaries to form perivascular endfeet at the blood–brain barrier [44]. Synapses in close proximity to astrocytic endfeet are reportedly considered highly stable [45]. Miñambres et al. [39] reported that astrocytes exposed to ethanol became apoptotic by anoikis and that the RhoA/ROCK-I/MLC pathway was involved in the mechanism of the onset of ethanol-induced apoptosis in astrocytes. Ma et al. [40] reported that Bcl-2 inhibitor of transcription-1 (Bit1), a regulator of anoikis, is expressed in the retina and is associated with anoikis in astrocytes and that β2-crystallin in the lens, a promoter of anoikis, is expressed in astrocytes. However, our review of the published literature yielded no descriptions of associations between Müller cells and anoikis.

Moreover, Müller cells express vimentin, which is the marker of mesenchymal cells [46]. Reportedly, mesenchymal cells are usually anoikis resistant [47], which indicates that DONFL is not caused by ILM peeling-associated damage to the Müller cells.

### 3.3. β. A3/A1-Crystallin and Anoikis

Crystallins are known to be major proteins in the lens. Mammalian lens crystallins are divided into alpha, beta, and gamma families in the order of molecular weight and are further classified into many subtypes. Of these, βA3/A1-crystallin is reportedly involved in the development and differentiation of the lens [48] and reportedly maintains the physiological functions of the lens [45]. Age-related degradation of the βA3/A1-crystallin gene causes cataracts [49,50,51], and mutations in this gene are known to cause cataracts [52,53]. βA3/A1-crystallin is also known as an anoikis-related protein. It has been reported that this crystallin is expressed in astrocytes, pericytes, ganglion cells, and RPE cells of the retina and optic disc, as ocular tissues other than the lens, and that it is also involved in the development and differentiation of these tissues [54,55]. However, there have been no published studies reporting that Müller cells express βA3/A1-crystallin.

βA3/A1-crystallin is a protein involved in the hydrolysis of lysosomes. It has been pointed out that if this function in the RPE deteriorates due to aging, waste materials, such as lipofuscin, accumulate in the RPE and cause AMD [56,57,58,59,60,61]. Moreover, the findings in another study have shown that a mutation in the βA3/A1-crystallin gene decreases the phagocytic capacity of RPE cells [62].

Reportedly, βA3/A1-crystallin is an important protein for maintaining the functions of astrocytes in the optic nerve head [63] and is essential for normal development and remodeling of retinal blood vessels [64,65]. Impairment of βA3/A1-crystallin in optic nerve head astrocytes is known to lead to impaired formation of blood vessels in the optic nerve head [66] and persistent hyperplastic primary vitreous [67].

### 3.4. E-Cadherin and Anoikis

Cadherins are a class of transmembrane glycoproteins involved in cell adhesion, and they play an important role in maintaining normal tissue morphogenesis [68]. Epithelial cadherin (E-cadherin) is one of the classical cadherins (along with N-, P-, R-, and B-cadherin) in the cadherin superfamily and is reportedly expressed on the cell surface of most epithelial tissues [69]. E-cadherin is involved in the regulation mechanism of epithelial cell proliferation, differentiation, and survival and is also known to be an anoikis-related protein [70,71]. N-cadherin maintains the structure of the ventricular zone by controlling adhesion between neural progenitor cells [72]. N-cadherin is necessary for neurons to adhere to radial processes, and it plays an important role in neuronal migration. In the process of neural differentiation, neurons lose E-cadherin and express N-cadherin [73]. Epithelial cells expressing N-cadherin are known to have mesenchymal properties, thus losing epithelial properties, which makes apoptosis less likely [74]. It has been reported that the loss of E-cadherin in RPE cells with AMD leads to epithelial–mesenchymal transition and that apoptosis is less likely to occur, thus resulting in the formation of connective tissue under the retina [55].

### 3.5. Anoikis and Retinal Ganglion Cells

As mentioned above, βA3/A1-crystallin, an anoikis-related protein, is not expressed in Müller cells, yet is expressed in retinal ganglion cells [55,56] Therefore, changes in the inner retinal layer observed in DONFL appearance is thought to be due to ILM peeling-induced anoikis in retinal ganglion cells and not in Müller cells. The retina is part of the central nervous system, and neurons constituting the retina normally express N-cadherin [75], while ganglion cells during embryogenesis express E-cadherin [76]. This means that relatively newly formed ganglion cells express E-cadherin and may undergo anoikis, whereas old ganglion cells originally present around the fovea centralis do not express E-cadherin and are, therefore, unlikely to undergo anoikis; accordingly, inner retinal changes featuring arcuate striae may occur.

### 3.6. Neurogenesis in the Macular Region

Although the fovea centralis constantly collects light and is continually stressed by light, its morphology and function are maintained without cell depletion throughout the life cycle of the cell. The fovea centralis also has anatomical features of a tissue stem cell niche, i.e., avascular, hypoxic, and concave. Based on the hypothesis that undifferentiated cells, such as retinal stem cells, are present in the fovea centralis, and nerve regeneration may be occurring, we conducted an immunohistological study of monkey foveal retinas [77]. After preparing frozen tissue sections of the retina, including the foveal pit, immunostaining was performed for glial fibrillary acidic protein (GFAP), nestin, vimentin, neuron-specific class III β-tubulin (Tuj-1), arrestin 4, neurofilament, CD117, CD44, and Ki67, followed by fluorescence and/or confocal microscopy examinations. Immunostaining of the tissue sections enabled clear observation of strongly GFAP-positive cells that corresponded to the inner-half layer of the foveolar Müller cell cone. The surface layer of the foveal slope was partially contained with GFAP and vimentin. Tuj-1-positive cells were observed in the innermost layer of the foveolar retina, which spanned to the surrounding ganglion cell layer. Moreover, colocalization of Tuj-1 and GFAP was observed at the foveal pit. The coexpression of CD117 and CD44 was found in the interphotoreceptor matrix of the fovea. The foveolar cone stained positive for both nestin and arrestin 4. However, the photoreceptor layer outside of the foveola displayed weak staining for nestin. Colocalization of nestin and vimentin was observed in the inner half of the Henle layer, while colocalization of nestin and neurofilament was observed in the outer half, predominantly. Scattered Ki67-positive cells were observed in the cellular processes of the outer plexiform layer and the ganglion cell layer around the foveolar (Figure 3A,B). The Müller cell cone was divided into GFAP-strongly positive cells, presumably astrocytes, in the inner layer and nestin-positive/GFAP-weakly positive radial glia-like cells in the outer layer. These findings indicated that groups of such undifferentiated cells in the foveola might be involved in maintaining morphology and regeneration. Moreover, the precursor cells of TUJ-1-positive ganglion cells in the fovea started to divide, playing a role in homeostatic regeneration in the macular region. The pathogenesis of an idiopathic MH is tangential vitreomacular traction, which is generated by a detached perifoveal posterior vitreous membrane. There is some possibility that an MH might be related to anoikis of these undifferentiated cells.

It is generally accepted that apoptosis occurs in a newly formed cell but is unlikely to occur in a senescent cell. In the adult hippocampus, most of the newly generated cells are eliminated by apoptosis, possibly because of their failure to integrate properly into neural networks [35]. Among the thousands of new neurons that integrate into the adult olfactory bulb each day, 50% are eliminated through apoptosis [78]. The continuous generation of new neurons in the adult brain is counterbalanced by accompanying cell death in the same regions [79]. As stated above, DONFL appearance is considered to be due to anoikis in a neuropoietic area around the fovea centralis; i.e., DONFL appearance may be due to anoikis that takes place when retinal stem cells present in the fovea centralis differentiate into ganglion cells. This might be the reason why retinal function was somewhat maintained after DONFL appearance occurred.

The findings in recent studies have demonstrated the presence of neural stem cells in the hippocampus and subventricular zone of the brain [80,81]. Neuroblasts migrate from the subventricular zone to the olfactory bulb along an astrocyte tube termed the ‘rostral migratory stream’ [82,83,84]. DONFL appearance is likely to be defects caused by anoikis in neuronal chains migrating along astrocytes in the Müller cell cone when neurons lose scaffolds for chain migration due to ILM peeling.

## 4. Reasons for the Frequent Occurrence of DONFL after Vitrectomy for an MH

It has been reported that DONFL frequently occurs after a vitrectomy for an MH [6,7,13,85]. We previously reported that the intravitreal activities of chymase and tryptase, which are serine proteases produced by mast cells, were increased in idiopathic MH and idiopathic ERM patients [86]. Based on the findings of our histological study, we also identified that mast cells residing in the bursa premacularis, a peculiar sac-like structure present in the premacular vitreous, might be the source of these serine proteases [87]. Mast cells are immune cells of the myeloid lineage that are present in mucosal and connective tissues throughout the body [88]. There are two types of mast cells, i.e., tryptase-positive mucosal mast cells and tryptase/chymase-positive connective tissue mast cells [89]. Chymase and tryptase, which are mast cell serine proteases, are involved in various physiological and pathological processes, including fibrosis, angiogenesis, and apoptosis, as well as tissue remodeling [90,91,92,93].

Chymase has apoptotic effects on various cells and reportedly degrades basement membranes by activating matrix metalloproteinase-9 [92]. In allergic conjunctivitis, anoikis reportedly occurs in conjunctival epithelial cells due to the apoptotic effect of chymase [94]. Originally, chymase, which is highly active in the vitreous, weakens the ILM. Furthermore, ILM peeling during vitrectomy may lead to anoikis-mediated apoptosis in ganglion cells.

In clinical practice, close long-term follow-up of patients with DONFL appearance and the application of recently advanced technologies, such as adaptive optics retinal imaging, are both considered useful [95]. The effect of tropomyosin receptor kinase B (TrkB), a neurotrophic factor receptor, on inhibiting anoikis may also be examined as a prophylactic measure against DONFL appearance [96,97]. We previously reported that as well as neurotrophic factors, such as nerve growth factor and brain-derived neurotrophic factor, neurotrophic factor receptors, such as tropomyosin receptor kinase A and TrkB, are expressed in cultured Müller cells [98]. Thus, a successful eliciting of these effects may lead to the prophylaxis of DONFL appearance.

## 5. Conclusions

As mentioned above, DONFL appearance may be changes caused by anoikis in ganglion cells associated with ILM peeling. However, further research is needed to verify this hypothesis.

## Figures and Tables

**Figure 1 ijms-22-01724-f001:**
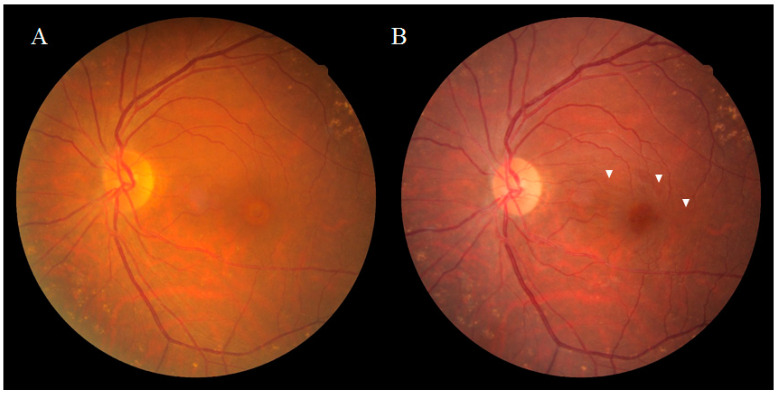
Fundus photograph of a patient obtained pre (**A**) and post (**B**) vitrectomy surgery for an idiopathic macular hole (MH). Ophthalmoscopically, the dissociated optic nerve fiber layer (DONFL) appearance features arcuate striae color changes, which appear to be multilinear shaped, that are frequently observed in the macular nerve fiber layer ((**B**), white arrowheads).

**Figure 2 ijms-22-01724-f002:**
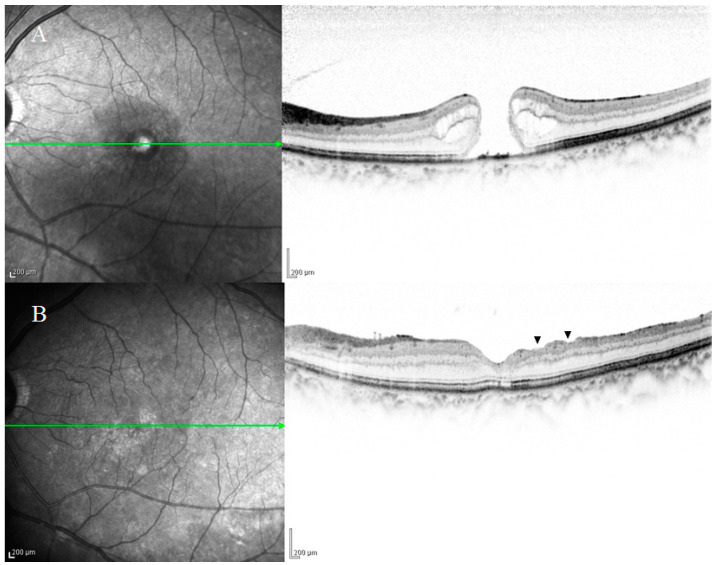
Optical coherence tomography (OCT) images of a patient with an idiopathic macular hole (MH) obtained pre-surgery (**A**) and at 6-months postoperative (**B**). OCT scans frequently showed thinning and irregularity of the inner retinal layer ((**B**), black arrowheads), thus indicating that the nerve fiber layer and ganglion cells were highly likely to have been damaged. The patient’s visual acuity was 20/100 before surgery and 20/25 after surgery.

**Figure 3 ijms-22-01724-f003:**
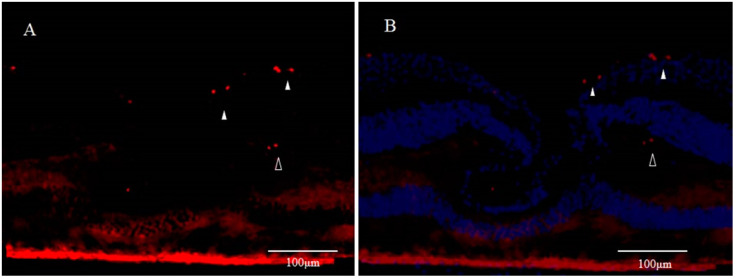
Immunostaining for Ki67 of Monkey Foveal Retina (cited from Reference [77]); (**A**) Scattered Ki67-positive spots (red) are visible in the innermost retinal layer in the parafoveola (white arrowheads). Moreover, some parts of the region believed to be the inner granular layer are also positive (unfilled arrowheads); (**B**) Immunostaining for Ki67 includes the immunostaining for DAPI (blue).

## Data Availability

The data presented in this study are available on request from the corresponding author.

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
