# Peer review of "Involvement of Anoikis in Dissociated Optic Nerve Fiber Layer Appearance"

_ijms, 2021, doi:10.3390/ijms22041724_

Round 1
Reviewer 1 Report
I congratulate the authors for their present work. However, there are few comments to improve the manuscript:
- Adding information on the incidence of DONFL may be helpful to the readers.
- Page 2, line 37: “Since the appearance of DONFL is not common in patients who have undergone vitrectomy without ILM peeling,” Please cite any reference wherein DONFL was reported post PPV without any ILM peeling.
- Figure 2 in the postop period also shows rarefaction of ellipsoid layer at the subfoveal level. On the other hand, DONFL appears parafoveal or perifoveally. In eyes with macular hole with loss or damage to EZ, its difficult to attribute lack of improvement in visual acuity to DONFL alone. Include the pre and postop visual acuity in the figure legend.
- Please mention the postop duration of OCT scans. Was it 1-2 months or 6-months post-surgery?
- What about the role of dyes such as ICG or brilliant blue i.e., concentration and the duration of staining on the appearance of DONFL?
- Please do mention this aspect in the discussion: “The role ofretinal side cellular debris present on the ILM after peeling and the extent of DONFL after surgery.” Here is one of the references. Steel et al. The relationship between a dissociated optic nerve fibre layer appearance after macular hole surgery and Muller cell debris on peeled internal limiting membrane Acta Ophthalmol. 2017 Mar;95(2):153-157. doi: 10.1111/aos.13195.
- Literature review also reveals doubts regarding the name DONFL. The word “dissociated” is not uniformly agreed. Please discuss the merits of these proposals and implications of other names such as inner retinal dimpling as proposed by Spaide. (DOI: 1097/IAE.0b013e3182671191)
- Does the differential density of Müller cells (such as between nerve fiber bundles, as compared with regions where there are fewer Müller cells) play a role in determining the dimension or location of DONFL? Please include this in the discussion.
Author Response
Reviewer 1
I congratulate the authors for their present work. However, there are few comments to improve the manuscript:
1.Adding information on the incidence of DONFL may be helpful to the readers.
Response to Comment 1: We greatly appreciate the Reviewer’s instructive comment. In accordance with the Reviewer’s cogent suggestion, please note that we have now added the following description about the incidence of DONFL:
Although the incidence of DONFL varies from report to report, it apparently occurs in approximately 30-60% of MHs in which an ILM peel is performed [1-6]. (Page 2, Lines 14-16)
2.Page 2, line 37: “Since the appearance of DONFL is not common in patients who have undergone vitrectomy without ILM peeling,” Please cite any reference wherein DONFL was reported post PPV without any ILM peeling.
Response to Comment 2: We greatly appreciate the Reviewer’s instructive comment. Please note that we have now cited the associated references, as follows:
Since the appearance of DONFL is not common in patients who have undergone vitrectomy without ILM peeling [2,3], it is thought that ILM peeling is certainly involved in its occurrence. (Page 2, Lines 9-12)
3.Figure 2 in the postop period also shows rarefaction of ellipsoid layer at the subfoveal level. On the other hand, DONFL appears parafoveal or perifoveally. In eyes with macular hole with loss or damage to EZ, its difficult to attribute lack of improvement in visual acuity to DONFL alone. Include the pre and postop visual acuity in the figure legend.
Response to Comment 3: We greatly appreciate the Reviewer’s instructive comment. Please note that we have now added the pre and postoperative visual acuity in the Legend of Figure 2, as follows:
The patient's visual acuity was 20/100 before surgery and 20/25 after surgery.
(Legend of Figure 2; Page 4, Line 4)
4.Please mention the postop duration of OCT scans. Was it 1-2 months or 6-months post-surgery?
Response to Comment 4: We greatly appreciate the Reviewer’s comment. Please note that we have now revised the explanation of the duration of OCT scans in the Legend of Figure 2, as follows:
Optical coherence tomography (OCT) images of a patient with an idiopathic MH obtained pre surgery (A) and at 6-months postoperative (B). (Legend of Figure 2; Page 4, Lines 1 and 2)
5.What about the role of dyes such as ICG or brilliant blue i.e., concentration and the duration of staining on the appearance of DONFL?
Response to Comment 5: We greatly appreciate the Reviewer’s helpful comment. Please note that we have now added an explanation in the Discussion section, and the associated reference in the References section, about the role of dyes such as ICG.
... while Park et al. [17] found no difference in the frequency of DONFL on the use of indocyanine green. (Page 4, Lines 9-21)
- Park, S.H.; Kim, Y.J.; Lee, S.J. Incidence of and risk factors for dissociated optic nerve fiber layer after epiretinal membrane surgery. Retina 2016, 36, 1469-1473; doi: 10.1097/IAE.0000000000000949. (Page 11, Lines 38 and 39)
6.Please do mention this aspect in the discussion: “The role of retinal side cellular debris present on the ILM after peeling and the extent of DONFL after surgery.” Here is one of the references. Steel et al. The relationship between a dissociated optic nerve fibre layer appearance after macular hole surgery and Muller cell debris on peeled internal limiting membrane Acta Ophthalmol. 2017 Mar;95(2):153-157. doi: 10.1111/aos.13195.
Response to Comment 6: We greatly appreciate the Reviewer’s instructive comment. In accordance with the Reviewer’s request, please note that we have now added the following explanation in the Discussion section, and the associated reference in the References section:
Steel et al. [16] described that the extent of DONFL observed postoperatively can be partly explained by the amount of cellular debris on the retinal side of the peeled ILM, ... (Page 4, Lines 17-19)
- Steel, D.H.W.; Dinah, C.; White, K.; Avery, P.I. The relationship between a dissociated optic nerve fibre layer appearance after macular hole surgery and Muller cell debris on peeled internal limiting membrane. Acta. Ophthalmol. 2017, 95, 153-157; doi: 10.1111/aos.13195. (Page 11, Lines 35-37)
7.Literature review also reveals doubts regarding the name DONFL. The word “dissociated” is not uniformly agreed. Please discuss the merits of these proposals and implications of other names such as inner retinal dimpling as proposed by Spaide. (DOI: 1097/IAE.0b013e3182671191)
Response to Comment 7: We greatly appreciate the Reviewer’s instructive comment. Please note that we have now added the discussion regarding the name DONFL as follows:
... and also described that DONFL seems to be the result of an interplay between trauma and healing processes constrained by NFL and do not appear to be due to dissociation of optic nerve fibers. As Spaide noted, the word "dissociated" may not be correct, and the term "inner retinal dimpling" may be more appropriate. (Page 6, Lines 6-9)
8.Does the differential density of Müller cells (such as between nerve fiber bundles, as compared with regions where there are fewer Müller cells) play a role in determining the dimension or location of DONFL? Please include this in the discussion.
Response to Comment 8: We greatly appreciate the Reviewer’s instructive comment. In section "3.6. Anoikis and Retinal Ganglion Cells", we speculated that Müller cells are not related to DONFL formation. In accordance with the Reviewer’s comment, please note that we have now added the following statement regarding the involvement of Müller cells in DONFL:
Therefore, changes in the inner retinal layer observed in DONFL appearance is thought to be due to ILM peeling-induced anoikis in retinal ganglion cells and not in Müller cells. (Page 7, Lines 47-49)

Reviewer 2 Report
This review article describes possible association between dissociated optic nerve fibers layer (DONFL) appearance and anoikis - a atype of apoptosis in eyes after vitreoretinal surgery due to macular hole. This is well written paper describing mechanisms involved in DONFL fundus appearance after ILM peeling. I have only one comment:
Line 140 –“There are two types of cells” – where? in the body? In the retina? what are kind of cells is described? Please specify
Author Response
Reviewer 2
This review article describes possible association between dissociated optic nerve fibers layer (DONFL) appearance and anoikis - a atype of apoptosis in eyes after vitreoretinal surgery due to macular hole. This is well written paper describing mechanisms involved in DONFL fundus appearance after ILM peeling. I have only one comment:
Line 140 –“There are two types of cells” – where? in the body? In the retina? what are kind of cells is described? Please specify
Response to Comment: We greatly appreciate the Reviewer’s instructive comment. The word "body" means the whole body. In accordance with the Reviewer’s comment, please note that we have now added the following explanations:
Anchorage-dependent cells proliferate while adhering to the culture vessel, and many cells such as epithelial cells belong to this category. On the other hand, cultured cells that proliferate while floating in the medium are known as anchorage-independent cells, such as blood cells. (Page 5, Lines 24-27)
The shape of the whole body is maintained by adhesion between cells that constitute the whole body, and these cells are anchorage dependent. Blood cells dispersed throughout the whole body are anchorage independent. (Page 5, Lines 27-29)
